# A Novel Deep Learning Approach to 5G CSI/Geomagnetism/VIO Fused Indoor Localization

**DOI:** 10.3390/s23031311

**Published:** 2023-01-23

**Authors:** Chaoyong Yang, Zhenhao Cheng, Xiaoxue Jia, Letian Zhang, Linyang Li, Dongqing Zhao

**Affiliations:** Institute of Geospatial Information, Information Engineering University, Zhengzhou 450001, China

**Keywords:** visual inertial odometry (VIO), CSI, geomagnetism, error state extended Kalman filter (ES-EKF), indoor localization

## Abstract

For positioning tasks of mobile robots in indoor environments, the emerging positioning technique based on visual inertial odometry (VIO) is heavily influenced by light and suffers from cumulative errors, which cannot meet the requirements of long-term navigation and positioning. In contrast, positioning techniques that rely on indoor signal sources such as 5G and geomagnetism can provide drift-free global positioning results, but their overall positioning accuracy is low. In order to obtain higher precision and more reliable positioning, this paper proposes a fused 5G/geomagnetism/VIO indoor localization method. Firstly, the error back propagation neural network (BPNN) model is used to fuse 5G and geomagnetic signals to obtain more reliable global positioning results; secondly, the conversion relationship from VIO local positioning results to the global coordinate system is established through the least squares principle; and finally, a fused 5G/geomagnetism/VIO localization system based on the error state extended Kalman filter (ES-EKF) is constructed. The experimental results show that the 5G/geomagnetism fusion localization method overcomes the problem of low accuracy of single sensor localization and can provide more accurate global localization results. Additionally, after fusing the local and global positioning results, the average positioning error of the mobile robot in the two scenarios is 0.61 m and 0.72 m. Compared with the VINS-mono algorithm, our approach improves the average positioning accuracy in indoor environments by 69.0% and 67.2%, respectively.

## 1. Introduction

With the development of science and technology, mobile robots have been introduced and are widely used in areas such as indoor assistance and environmental awareness. In outdoor areas, the global navigation satellite system (GNSS) can provide an accurate and reliable real-time positioning signal source for robots, meeting the location service needs in most scenarios. However, in indoor environments, GNSS signals are obscured by walls and cannot provide location information to the robot, which requires the use of other sensors to achieve positioning.

In recent years, with the development of computer vision technology, simultaneous localization and mapping (SLAM) based in cameras has been developed rapidly, where visual inertial odometry (VIO) uses measurements from vision sensors and inertial measurement units (IMU) to determine the relative motion of a carrier [1,2]. The camera can capture rich environmental information, while the IMU can provide accurate position estimation in a short period of time, and both have the characteristics of small size and low power consumption. VIO has gradually developed into one of the main methods for indoor positioning of robots. VIO can achieve desirable accuracy when the light conditions and image quality are good, but it lacks a global reference of position information, and is in essence a local positioning algorithm [3]. During indoor positioning of long trajectories, VIO suffers from cumulative errors, resulting in drift in position estimation.

Compared to the VIO local positioning technique, Wi-Fi positioning [4], the fifth generation mobile communication network (5G) positioning [5] and geomagnetic matching [6] have the advantage of global positioning and can provide absolute position information. Each of their observations is independent, errors do not accumulate and global positioning results do not drift. However, the positioning frequency and position accuracy of these methods are low, so they cannot meet the needs of high precision position applications [7]. Therefore, combining VIO with high local accuracy but accumulated errors and a global positioning technique with low local accuracy but no drift can compensate for each other [8].

Due to the complex indoor environment, any single global positioning signal source will have limitations and low position accuracy. For example, geomagnetic matching uses geomagnetic sequences as fingerprint features, and considering the impact of robot speed variation on time sequences, the dynamic time warping (DTW) algorithm is usually applied to match positioning [9], but the similarity of geomagnetic sequence features degrades the matching accuracy. With the large-scale popularization and application of 5G, the localization methods based on 5G New Radio (NR) have attracted huge amount of attention [10]. Channel state information (CSI) can reflect the fine-grained signal propagation characteristics of wireless communication links, and it has been proven to favor indoor positioning methods based on 5G signals in the case of multiple subcarriers of CSI [11]. However, in dynamic scenarios, CSI is prone to signal buildup and is affected by signal fluctuations. Therefore, fusion of 5G and geomagnetic signals can reduce positioning errors and provide more reliable global positioning results.

The main contributions of this paper are as follows:

(a) An error back propagation neural network (BPNN)-based 5G/geomagnetic fusion localization method is proposed, which can provide reliable global localization results in indoor environments;

(b) Based on the least squares principle, we solve the transformation parameters of the VIO local positioning results to the global coordinate system, which enables the VIO to output global positioning results consistently;

(c) We fuse the global positioning results of 5G/geomagnetic with the local positioning results of VIO by means of the error state extended Kalman filter (ES-EKF) to obtain more stable combined positioning results;

(d) Finally, we build an experimental platform based on existing 5G devices, magnetometers and cameras, and demonstrate the reliability and stability of the fusion method under different scenarios.

The rest of the paper is organized as follows: Section 2 reviews existing research on 5G, geomagnetism and VIO localization. Section 3 introduces our overall fusion framework and the fusion localization methods. Section 4 presents our experimental environment and experimental results. Finally, the fifth part summarizes the direction of our future work.

## 2. Related Works

With the development of science and technology, mobile intelligent terminals can provide a variety of service modes for people indoors, and the prerequisite for intelligent terminals to provide services is the need to obtain real-time and accurate location information about themselves [12]. Research scholars have developed a variety of indoor positioning technologies that can provide services for technologies, such as the Internet of Things (IoT) navigation. At present, the most commonly used indoor positioning techniques include three main categories. One is geometric relationship localization based on reference points, such as time-of-arrival-based localization [13] and time-of-arrival difference-based localization [14], but such localization can only provide services in line-of-sight (LOS) scenarios, and non-line-of-sight (NLOS) has a greater impact on localization accuracy. The second is the positioning method based on heading projection, such as inertial navigation [15], which has a high positioning accuracy at short distances, but has cumulative errors. Third, location fingerprint positioning relies on existing fingerprint information for positioning [16], such as the received signal strength (RSS) and CSI of communication networks. However, existing indoor positioning techniques have low accuracy and often have limited output frequency, making it difficult to provide real-time and reliable positioning results. In contrast, the positioning accuracy of VIOs is high in short periods of time, but there are cumulative errors that do not guarantee long duration applications. As a result, research scholars have begun to focus on the integration of VIO with other indoor positioning technologies for positioning.

In recent years, with the research on monocular visual odometry, many classical frameworks have been born, such as LSD-SLAM [17], ORB-SLAM [18], SVO [19], etc. However, these monocular odometry methods lead to scale ambiguity of the system and poor robustness of the algorithm. To solve this problem, researchers have proposed joint camera and IMU position estimation methods. In 2015, Bloesch [20] proposed a monocular visual inertial odometry called ROVIO, which fuses IMU and image information through iterative extended Kalman filtering, but as the method does not contain a loop closure, localization errors accumulate indefinitely. OKVIS [21] is a VIO based on keyframe optimization, which optimizes the nearest state by marginalizing past states and measurements over a sliding window of fixed size, but its keyframes are selected based on spacing and do not take into account poses in continuous time. VINS-mono [22,23,24,25] translates local state estimation via a sliding window into a graph-based nonlinear least squares problem, which has a closed-loop detection and repositioning mechanism, it is currently one of the better monocular visual inertial odometers. By adding IMU information, ORB-SLAM was extended in 2020 to a visual inertial SLAM, namely ORB-SLAM3 [26], and fuses visual and inertial navigation information through a non-linear optimization approach.

Two key issues in practical applications of VIO are the alignment of local position estimates to the global coordinate system and the reduction in cumulative positioning errors. He [27] proposed a systematic approach to extend VINS-mono to provide global state estimates for outdoor vehicle navigation by fusing VIO local position estimates with GNSS information, and Kang [28] aligned VIO with a vehicle dynamics model tightly coupled to achieve meter-level localization results for dynamic outdoor scenes and underground car parks. All the above studies focus on the localization effect of large carriers, such as vehicles in outdoor scenes, and are not applicable to small carriers such as indoor unmanned aerial vehicles and unmanned vehicles. Zhang [29] optimized the position estimation of VIO by adding range information by installing UWB tags on multiple robots, but the output was still in the local coordinate system and the absolute position output was not considered. Gao [30] proposed a UWB-assisted low-drift VIO indoor localization method, which reduced the drift of VIO and improved the robustness of motion tracking by adding anchor position and ranging information to the VIO tightly coupled frame. UWB, as a range-based localization method, is susceptible to multipath effects and non-visual range errors and it is difficult to provide stable and reliable position information in indoor environments.

With the development of indoor positioning technology, an indoor positioning method based on location fingerprint has received wide attention. This method regards the unique feature information of each coordinate point as a fingerprint and achieves the positioning effect by matching the feature information of the point to be measured with the data of the fingerprint library. The most commonly used feature information includes magnetic field strength, received signal strength (RSS) of wireless signal, CSI, etc. [31] proposed an indoor positioning system based on a Bayesian graphical model, which realized three-dimensional positioning of an indoor environment through radio frequency (RF) fingerprint technology, and its overall positioning error was about 2.9 m. Qin [32] proposed a path matching-based indoor geomagnetic positioning system combing DTW and particle filtering algorithms to achieve positioning with fingerprint matching, and the positioning accuracy was within 1 m. Xu [33] used the DTW algorithm for fingerprint matching based on the indoor geomagnetic and barometric signal characteristics to achieve high-precision positioning on multiple floors indoors. The geomagnetic matching approach can provide drift-free global position estimation, which can provide absolute position coordinates for the VIO and reduce the accumulated positioning error of the VIO over a long period of time. Liu [34] proposed a tightly coupled algorithm for VIO and magnetometer fusion, which determined the pose through gravitational and geomagnetic vectors, constructed geomagnetic vector residuals using pose quaternions, and fused them into a graph optimization algorithm to achieve high accuracy positioning of the unmanned vehicles.

Due to the single feature information, the localization based on geomagnetic matching is prone to mismatching in large-area scenarios, and researchers usually fuse other sensors to achieve indoor global localization. Pan [35] proposed a combined Wi-Fi/geomagnetism localization method, which used Wi-Fi for preliminary localization and used the preliminary localization results to filter the geomagnetic localization area. Then, geomagnetic matching localization was performed within the area, effectively reducing the mismatching rate. Basmag [36] is an indoor positioning system based on the hidden Markov model (HMM), which converts geomagnetic vectors into fingerprint sequences by pedestrian heading projection (PDR) to obtain more accurate conversion probabilities in HMM, and the system is effective and robust for users with different walking styles. With the large-scale use of 5G base stations, many scholars have researched 5G CSI-based localization. Gao [37] generated 5G CSI through a simulation approach and used ray-tracing channel models to achieve indoor and outdoor localization, but the effectiveness of the method has not been verified in real-world scenarios. Hi-Loc [11] realized indoor positioning based on a convolutional neural network and long and short term memory through independently studied 5G CSI transceiver equipment, and achieved meter-level positioning accuracy. Kia [38] used real 5G CSI data based on ray tracing to achieve intercity fingerprint matching localization by means of a convolutional neural network algorithm. In this context, this paper fuses 5G CSI and geomagnetic vectors to obtain a more reliable global localization result, and then fuses this result with the local localization result output from VIO to obtain the final fused localization result.

## 3. Proposed Method

Figure 1 illustrates the overall architecture of the fusion localization method used in this paper. Firstly, the image, acceleration, pose angle, geomagnetic intensity and CSI information are obtained from the camera, IMU, magnetometer and 5G device, respectively; the camera and IMU information outputs the local position estimation through the tightly coupled VIO; and the 5G CSI fuses the geomagnetic data to output the global position estimation. Then, the local position estimate output from the VIO is aligned to the global coordinate system. Finally, the two position estimates are fused based on the ES-EKF. It should be noted that since all the localization methods used in this paper are localized in a planar right-angle coordinate system based on a position fingerprint library, we only consider the transformation relationship from the local coordinate system of the VIO to this coordinate system. Of course, our fused localization methods can be extended to 3D space to support different global coordinate systems.

Our method consists of four parts, the first part is pre-processing of 5G and geomagnetic signals, the second part is a global localization method based on 5G/geomagnetism fusion, the third part is VIO local position estimation and coordinate conversion, and the final part is global position estimation based on ES-EKF.

### 3.1. Signal Preprocessing

In the positioning based on 5G NR, CSI generally refers to the channel state information obtained by demodulation at the receiving terminal of the 5G signal, which mainly includes the signal amplitude, phase and time delay. As the information on multipath impact, signal propagation time and phase change are different at different locations, but remain relatively stable over time at the same location, there is a certain mapping relationship between location and CSI. Figure 2 depicts the images of 5G NR CSI data from two different locations. Therefore, the change in CSI in the 5G frequency domain for multiple carriers can be used to characterize the amplitude energy change, phase change and multipath state of the signals received at different locations. CSI includes channel state information for each subcarrier in the frequency domain, and the frequency domain model of the channel state is described as
(1)Y=HX+N
where Y, H, X and N represent the received signal vector, the channel matrix, the transmitted signal vector and the Gaussian white noise, respectively. The channel matrix describes the fading factor of the signal on each transmission path, where the value of each element contains information on signal scattering, environmental fading and distance attenuation. The channel matrix is described as
(2)$$H_{i}=\left|H_{i}\right|e^{j\sin\left(∠H_{i}\right)}$$
where $\left|H_{i}\right|$ and $∠H_{i}$ denote the amplitude response and phase response, respectively. Each CSI in this paper contains 60 subcarriers, namely i∈[1,60].

As can be seen from Figure 2, the amplitudes of the 60 subcarriers at the same position remain relatively stable over a period of time, while they show different patterns in variation at different positions. In order to solve the problem of overfitting of the neural network caused by the large amount of data, and to reduce the computations of the neural network learning training and matching and improve the efficiency, we downscaled the CSI amplitudes of the 60 subcarriers. Retaining the most important variation regularity features of the CSI amplitude, we took the weighted average of the data of 60 subcarriers, and the weighting factor was chosen based on a free-space propagation loss model. In free space, let the transmit power be $P_{t}$ and the receive power be $P_{r}$. Then,
(3)$$P_{r}=\frac{A_{r}}{{4\pi d}^{2}}P_{t}G_{t}$$
where $A_{r}=\lambda^{2}\times G_{r}/4\pi$, λ is the wavelength, $G_{r}$ and $G_{t}$ are the transmitting and receiving antenna gains, respectively, and d is the distance between the transmitting and receiving antenna. The propagation loss, L, in free space is defined as
(4)$$L=P_{t}/P_{r}.$$

When $G_{r}=G_{t}=1$, the propagation loss in free space is
(5)$$L={\left(\frac{4\pi d}{\lambda }\right)}^{2}.$$

If expressed in decibels (dB), then
(6)$$L=22.7+26\mathrm{lg}f+36.7\mathrm{lg}d-0.3\left(h_{\mu t}-1.5\right)$$
where f is the center frequency of each subcarrier and $h_{\mu t}$ is the absolute height of the receiver [39]. Due to $h_{\mu t}$ and the distance d between the transmitting and receiving antennas at the same location remaining constant, the weighting factor is only related to the center frequency of each subcarrier under the free-space propagation loss model. After the subcarrier aggregation process, the CSI data on 60 subcarriers are downscaled into one set of CSI data (Figure 3), which can greatly reduce the time of machine learning operations and retain the most important CSI amplitude change trend characteristics, improving the operational efficiency of the indoor positioning system.

Assuming that the magnetometer measures the magnetic field strength in each of the three axes as $m_{x},m_{y},m_{z}$, the geomagnetic modulus is
(7)$$m=\sqrt{m_{x}^{2}+m_{y}^{2}+m_{z}^{2}}.$$

Since $m_{x},m_{y},m_{z}$ are the three-axis magnetic field components with the magnetometer itself as the coordinate system, as shown in Figure 4a, when the magnetometer is in the same position holding a different pose, its readings will change significantly and only the geomagnetic modulus value remains relatively stable. In order to overcome the interference of pose changes on the localization process, Wu [40], Liu [41] and others used only the geomagnetic modulus to perform geomagnetic sequence matching, but the lack of features easily caused mismatching. To increase the geomagnetic features, we ensure the relative stability of the geomagnetic data by rotating the geomagnetic data to the navigation coordinate system.

Assuming that the magnetic field strength component measured by the magnetometer in its own carrier coordinate system for the three axes is $m^{b}=\left[m_{x}^{b},m_{y}^{b},m_{z}^{b}\right]$, and the magnetic field strength component in the rotated navigation coordinate system is $m^{n}=\left[m_{x}^{n},m_{y}^{n},m_{z}^{n}\right]$, then the equation for converting the magnetometer observation vector from the carrier coordinate system to the navigation coordinate system is
(8)$$m^{n}=C_{b}^{n}m^{b}$$
where $C_{b}^{n}$ is the directional cosine matrix, expressed as
(9)$$C_{b}^{n}=\left[\begin{array}{ccc} \cos\varphi \cos\psi & -\cos\theta \sin\psi +\sin\theta \sin\varphi \cos\psi & \sin\theta \sin\psi +\cos\theta \sin\varphi \cos\psi \\ \cos\varphi \sin\psi & \cos\theta \cos\psi +\sin\theta \sin\varphi \sin\psi & -\sin\theta \cos\psi +\cos\theta \sin\varphi \sin\psi \\ -\sin\varphi & \sin\theta \cos\varphi & \cos\theta \cos\varphi \end{array}\right]$$
where φ, θ and ψ are the pitch, roll and yaw angles measured by the magnetometer, respectively. The geomagnetic vector feature after rotation is shown in Figure 4b. After conversion, the geomagnetic feature is less influenced by the carrier pose at the same position and remains relatively stable. From this, we use the geomagnetic field component under the navigation system as the geomagnetic feature vector and fuse the 5G CSI data after subcarrier aggregation for position fingerprint matching localization.

### 3.2. Global Location Estimation Based on BPNN

The BPNN is a feedforward neural network trained according to the error back propagation algorithm, which has a strong nonlinear mapping capability and can meet the requirements of nonlinear mapping between CSI amplitude information, geomagnetic intensity information and real coordinates [42]. The learning and training process is shown in Figure 5.

The BPNN consists of an input layer, a hidden layer and an output layer, of which the input layer is responsible for receiving the 5G and geomagnetic information, the hidden layer is responsible for processing the received signal information and establishing a non-linear mapping relationship between the received signal and the coordinates, and the output layer is responsible for outputting the estimated location coordinates. When the input layer receives the 5G and geomagnetic data, it is passed to the hidden layer. The hidden layer will weight and non-linearly transform the data from each neuron in the input layer and pass it to the output layer by forward propagation, and the output layer will output the received results to the outside world. When the output does not match or differs significantly from the desired output, the weights and bias terms of the individual neurons are corrected by back propagation until the threshold of error or the maximum number of iterations is reached. When the network learning training is complete, the 5G and geomagnetic data of the point to be measured is input, and the localization results can be output by forward propagation.

Figure 6 shows the basic structure of a single neuron. During the processing of the neural network, each neuron performs a weighted summation and non-linear transformation of all the input signals from the previous layer of the network and then passes them to the next neuron.

Suppose the input vector of a neuron is $X={\left(x_{1},x_{2},\ldots ,x_{n}\right)}^{T}$, and $W={\left(w_{1},w_{2},\ldots ,w_{n}\right)}^{T}$ is the vector of weights of the input nodes connecting the nodes of the neuron, where n is the number of input neurons of the previous layer of the network. Then, the weighted input of this neuron node is
(10)$$s=\sum_{i=1}^{n}x_{i}w_{i}+\theta .$$

Then, the state output of this neuron is
(11)y=f(s)
where f(⋅) is the activation function, which constitutes the nonlinear transformation relationship between the input and output of neurons. The Sigmoid function can convert the input value to an output between 0 and 1, which can achieve fast convergence for a single hidden layer neural network, and we chose the Sigmoid function as the activation function of BPNN in this paper, and its expression is
(12)$$y=f(s)=\frac{1}{1+e^{-s}}.$$

### 3.3. Local Location Estimation Based on VIO and Coordinate Transformation

For local position estimation, we used existing VIO algorithms. There are many excellent VIO algorithms such as VINS-mono [24], ORB-SLAM3 [25], etc., mentioned in Section 2, any of which can output the local position estimation. In the experiments in this paper, we performed fusion localization using the VINS-mono algorithm. VINS-mono estimates the depth of the pose and features of several IMU frames within a sliding window. These states are defined as:(13)$$\begin{array}{c} X_{l}=\left[x_{0},x_{1},\ldots x_{n},\lambda_{0},\lambda_{1},\ldots \lambda_{m}\right]\\ x_{k}=\left[p_{b_{k}}^{l},v_{b_{k}}^{l},q_{b_{k}}^{l},b_{a},b_{g}\right],k\in [0,n] \end{array}$$
where the state $x_{k}$ of the $k^{\prime }th$ IMU consists of the position $p_{b_{k}}^{l}$, velocity $v_{b_{k}}^{l}$ and direction $q_{b_{k}}^{l}$ of the IMU center with respect to the local coordinate system, l. We use quaternions to represent the direction. The first IMU pose is set as the reference coordinate system, and $b_{a}$ and $b_{g}$ are the accelerometer bias and gyroscope bias, respectively. When first observed in the camera frame, the features are parameterized by the inverse depth, λ, and the pose estimation is formulated as a non-linear least squares problem.
(14)$$\min_{X_{l}}\left\{{\left\|r_{p}-H_{p}X\right\|}^{2}+\sum_{k\in B}{\left\|r_{B}\left({\hat{z}}_{b_{k+1}}^{b_{k}},X\right)\right\|}_{P_{b_{k+1}}^{b_{k}}}^{2}+\sum_{(l,j)\in C}\rho ({\left\|r_{C}\left({\hat{z}}_{l}^{c_{j}},X\right)\right\|}_{P_{l}^{c_{j}}}^{2})\right\}$$
where $r_{B}\left({\hat{z}}_{b_{k+1}}^{b_{k}},X\right)$ and $r_{C}\left({\hat{z}}_{l}^{c_{j}},X\right)$ denote the inertial and visual residuals, respectively, $\left[r_{p},H_{p}\right]$ includes information on past marginalization states and ρ(⋅) denotes the Huber robustness parametrization. The VIO implements real-time 6-degree-of-freedom pose estimation within a local frame, and this paper focuses only on the output position and velocity estimates.

The conversion of the local position estimate of VIO to the global coordinate system is shown in Figure 7. 

Assume that the output of a position is $loc_{l}=(x_{l},y_{l})$ in the local coordinate system and $loc_{g}=(x_{g},y_{g})$ in the global coordinate system, then the conversion formula for the local to global coordinates is
(15)$$loc_{l}=Rloc_{g}+T$$
where R and T represent the rotation matrix and translation matrix, respectively, where the specific relationship is
(16)$$R=\left[\begin{array}{cc} \cos\alpha & \sin\alpha \\ -\sin\alpha & \cos\alpha \end{array}\right],T=\left[\begin{array}{c} D_{x}\\ D_{y} \end{array}\right],$$
that is,
(17)$$\left[\begin{array}{c} x_{g}\\ y_{g} \end{array}\right]=\left[\begin{array}{c} D_{x}\\ D_{y} \end{array}\right]+\left[\begin{array}{cc} \cos\alpha & \sin\alpha \\ -\sin\alpha & \cos\alpha \end{array}\right]\times \left[\begin{array}{c} x_{l}\\ y_{l} \end{array}\right].$$

Each pair of point coordinates can be calculated from the above equation to obtain a set of corresponding $\left({x_{g}}^{\prime },{y_{g}}^{\prime }\right)$, treating their corresponding observations in the global coordinate system as true values, the following can be obtained:(18)$$\left[\begin{array}{c} x_{g}\\ y_{g} \end{array}\right]=\left[\begin{array}{c} V_{x}\\ V_{y} \end{array}\right]+\left[\begin{array}{c} {x_{g}}^{\prime }\\ {y_{g}}^{\prime } \end{array}\right].$$

Treating $D_{x}$, $D_{y}$ and α as unknowns and linearizing and expanding the above two equations, then
(19)$$\left[\begin{array}{c} V_{x}\\ V_{y} \end{array}\right]=\left[\begin{array}{ccc} 1 & 0 & \cos\alpha \cdot x_{l}+\sin\alpha \cdot y_{l}\\ 0 & 1 & -\sin\alpha \cdot x_{l}+\cos\alpha \cdot y_{l} \end{array}\right]\times \left[\begin{array}{c} D_{x}\\ D_{y}\\ d\alpha \end{array}\right]-\left[\begin{array}{c} x_{g}-{x_{g}}^{\prime }\\ y_{g}-{y_{g}}^{\prime } \end{array}\right].$$

The above equation is translated into matrix form as
(20)V=BX−L,
where
(21)$$X=\left[\begin{array}{c} D_{x}\\ D_{y}\\ d\alpha \end{array}\right]=\left[\begin{array}{c} T\\ R \end{array}\right].$$

The rotation and translation matrices can be solved by using the least squares principle, and the local position estimate is then converted to a global position estimate by Equation (15).

### 3.4. Global Location Estimation Based on ES-EKF

The ES-EKF-based fusion localization process is shown in Figure 8. 

In the fused localization algorithm, we use the planar coordinate system obtained by the 5G/geomagnetic fingerprint library as the global coordinate system, the position and velocity of the carrier as the state vector of the system, then the state vector at the $k^{\prime }th$ moment is
(22)$$X_{k}={\left(x_{k},y_{k},v_{k}^{x},v_{k}^{y}\right)}^{T},$$
where $\left(x_{k},y_{k}\right)$ represents the coordinates of the combined system in the two-dimensional plane global coordinate system at the $k^{\prime }th$ moment, and $v_{k}^{x},v_{k}^{y}$ represents the velocity of the combined system in the x component and y component at the $k^{\prime }th$ moment, respectively. Assuming that the valuation of the state vector is ${\hat{X}}_{k}$ at the $k^{\prime }th$ momentand the corresponding state estimation error is $\delta X_{k}$, namely $X_{k}={\hat{X}}_{k}+\delta X_{k}$, then the error state model of the system can be expressed as
(23)$$\left\{\begin{array}{c} \delta x_{k}=\delta x_{k-1}+\delta v_{k-1}^{x}t+\frac{t^{2}}{2}a_{k-1}^{x}\\ \delta y_{k}=\delta y_{k-1}+\delta v_{k-1}^{y}t+\frac{t^{2}}{2}a_{k-1}^{y}\\ \delta v_{k}^{x}=\delta v_{k-1}^{x}+a_{k-1}^{x}t\\ \delta v_{k}^{y}=\delta v_{k-1}^{y}+a_{k-1}^{y}t \end{array}\right.,$$
where t is the sampling interval time of the combined system and $a_{k-1}^{x}$ and $a_{k-1}^{y}$ represent the acceleration of the combined system in the x direction and y direction at the (k−1)th moment, respectively. We use the position and velocity of the VIO output as the estimated state vector of the combined system for error state update, then the prediction of the error state and covariance at moment, k, is
(24)$$\left\{\begin{array}{c} \delta X_{k|k-1}=F\delta X_{k-1}+W_{k-1}\\ P_{k|k-1}=FP_{k-1}F^{T}+Q_{k-1} \end{array}\right.,$$
where F is the state matrix of the system and $W_{k-1}$ is the process noise. $P_{k-1}$ is the optimal estimate of the error state covariance at the (k−1)th moment and $Q_{k-1}$ is the corresponding noise covariance. F and $W_{k-1}$ can be defined as
(25)$$F=\left(\begin{array}{cccc} 1 & 0 & t & 0\\ 0 & 1 & 0 & t\\ 0 & 0 & 1 & 0\\ 0 & 0 & 0 & 1 \end{array}\right),\;\;\;W_{k-1}=\left(\begin{array}{c} \frac{t^{2}}{2}a_{k-1}^{x}\\ \frac{t^{2}}{2}a_{k-1}^{y}\\ a_{k-1}^{x}t\\ a_{k-1}^{y}t \end{array}\right).$$

Since the state vector of the combined system includes only the position and velocity of the carrier, we treat the acceleration as the random noise of the system. For the combined system, we take the difference between the VIO and the 5G/geomagnetism output position estimate as the observation vector, $Y_{k}$, and the observation equation for the system is
(26)$$Y_{k}=H\delta X_{k}+V_{k}$$
where $V_{k}$ is the observed noise and $Y_{k}$ and H are expressed as
(27)$$Y_{k}=\left(\begin{array}{l} x_{k}^{5G/\mathrm{Mag}}-x_{k}^{\mathrm{VIO}}\\ y_{k}^{5G/\mathrm{Mag}}-y_{k}^{\mathrm{VIO}} \end{array}\right),\;\;\;H=\left(\begin{array}{llll} 1 & 0 & 0 & 0\\ 0 & 1 & 0 & 0 \end{array}\right),$$
where $x_{k}^{5G/\mathrm{Mag}}$ and $y_{k}^{5G/\mathrm{Mag}}$ are the position estimates of the 5G and geomagnetism fusion at the kth moment. The error states and covariance matrix can be updated by means of the state equations and observation equations as follows:(28)$$\left\{\begin{array}{c} K_{k}=P_{k|k-1}H^{T}{\left(HP_{k|k-1}H^{T}+R_{k}\right)}^{-1}\\ \delta X_{k|k}=\delta X_{k|k-1}+K_{k}\left(Y_{k}-H\delta X_{k|k-1}\right)\\ P_{k|k}=\left(I-K_{k}H\right)P_{k|k-1} \end{array}\right.$$
where $K_{k}$ is the Kalman gain matrix, $P_{k}$ is the observed covariance at kth moment and $\delta X_{k|k-1}$ and $P_{k|k-1}$ are the update of the error state and covariance matrix, respectively. The optimal position estimate at kth moment is obtained from $X_{k}={\hat{X}}_{k}+\delta X_{k}$.

## 4. Experimental Results and Analysis

In order to verify the indoor positioning performance of the proposed fusion positioning algorithm, we first evaluated the global positioning method with 5G and geomagnetism fusion, and then conducted field tests under two different scenarios. The experimental equipment was built as shown in Figure 9, and Table 1 shows the detailed parameters of the equipment used for the experiments.

We first used continuous path acquisition to collect 5G and geomagnetic information in the experimental area, and then interpolated the positions on both sides of the path by kriging interpolation, to make a uniform distribution of the location fingerprint library points with an interpolated resolution of 0.5 m × 0.5 m. After the location fingerprint library was established, we used a pre-built unmanned vehicle carrier platform to move along the established trajectory, with three sensor output measurements delivered simultaneously to the same computer, which were then solved in real time using our proposed fusion localization algorithm. The T265 camera was responsible for outputting image and IMU information at 50 Hz and 200 Hz, respectively, the magnetometer was responsible for outputting geomagnetic 3D vector information at 50 Hz and the 5G device was responsible for outputting CSI data, which was also at 50 Hz. In the VIO localization process, the IMU output was pre-integrated and then fused with the image data, its final output of position estimation data was also at 50 Hz. During data collection, 5G collects CSI for the single base station case, containing a total of 60 subcarrier amplitudes, which were localized as fingerprint information after subcarrier aggregation based on the free-space model in Section 3, Part A. The 3D vector intensity information collected by the magnetometer was rotated by the vectors in Section 3, Part A and then used as fingerprint information to be fused with 5G for localization. The two experimental scenarios are shown in Figure 10 and Figure 11, respectively.

### 4.1. 5G and Geomagnetic Positioning Performance

We first tested the localization effect of the 5G and geomagnetism fusion global localization method using some points in a conference room. The cumulative distribution function (CDF) of errors for the 5G localization, geomagnetic matching localization and 5G and geomagnetism fusion localization methods are given in Figure 12, and Table 2 shows a comparison of the localization errors of the three localization methods.

As can be seen from Table 2, the global positioning results based on single sensors are all poor, while the maximum positioning error of the 5G and geomagnetism fusion indoor positioning method proposed in this paper was 1.47 m and the average positioning error was 0.49 m. It can also be seen from Figure 12 that the positioning errors after fusion are mostly distributed within 1 m. Compared to Hi-Loc [11], our proposed algorithm does not differ much in localization accuracy in approximately the same scenario with a single 5G localization, but after fusion, our localization accuracy improvement over the Hi-Loc algorithm is about 24.6%. Therefore, our proposed 5G and geomagnetism fusion positioning method can make up for the single sensor deficiencies, improve indoor positioning accuracy and provide reliable global position estimation for subsequent local and global fusion positioning.

### 4.2. Conference Room

As the VIO localization method is susceptible to lighting problems and is also prone to cumulative errors, we first tested the localization effect of our localization algorithm in a poorly lit scene. We tested it at night in a conference room; the experimental area was about 58 square meters, and only one row of lights near the x-direction was kept on during the experiment, the rest of the lights were turned off. We drove the unmanned vehicle from point (0, 0) in a clockwise direction. Figure 13 shows the trajectories output by our fusion algorithm and VIO, respectively.

As can be seen from the figure, the VIO trajectories generally perform poorly and have large deviations from the true trajectories, which may be caused by the lack of indoor lighting. Whereas the output trajectory is relatively close to the true trajectory on the left-hand side, it gradually deviates from the true trajectory on the remaining three sides, mainly due to the cumulative errors that exist over time. In contrast, our proposed fusion localization algorithm largely suppresses the drift in position estimation, and after ES-EKF filtering, its trajectory output is smoother and closer to the true trajectory. However, after fusion, its predicted trajectory on the left side is somewhat far from the real trajectory, while the remaining three sides are significantly improved. This may be due to the larger fusion positioning error of 5G and geomagnetic relative to VIO, which caused a decrease in the positioning accuracy of the left side after fusion. Overall, its positioning trajectory has been greatly improved. The CDF plots of the two methods are shown in Figure 14.

The localization error distribution of VIO is more diffuse, mainly concentrated in the range of 0–1 m and 3–4 m, which is caused by its large error on the x-axis compared with Figure 13. The fused position estimation algorithm, on the other hand, is focused almost entirely in the 0–1 m range. Table 3 shows a comparison of the localization errors of the two algorithms. After fusion, the average localization error of our algorithm was 0.61 m, a 69.0% improvement over the single VINS-Mono [25] localization method. This shows that the fusion of global and local position estimation has largely overcome the problem of cumulative errors in local position estimation. Table 3 shows a comparison of the localization errors of the two algorithms.

### 4.3. Covered Corridor

Experiment (B) verified the effectiveness of our algorithm for localization in a small area in a dark night-time environment. To further validate the localization performance of the fusion localization algorithm, we tested it in a corridor. The experimental scene is shown in Figure 11, which covers an area of approximately 375 m^2^, a larger area than the conference room; however, the experiment was conducted in daylight and the light was brighter in experiment (A) than in experiment (B). Additionally, to verify the effect of the driving route on the results of the experiment, during the experiment we drove the unmanned vehicle still from the point (0, 0), but in a counterclockwise direction for one revolution. Additionally, a comparison of the trajectories estimated by the two algorithms with the real trajectories is shown in Figure 15.

The output trajectory was closer to the real trajectory in experiment (A) than experiment (B), because the experiment was carried out during the day with good light, which enabled more feature points to be extracted. It can be seen that the brightness of the light had a greater impact on the position estimation of the VIO, and therefore this experiment is more reflective of the real VIO position estimation because of the better light. The position estimation of the VIO is better in the early stage, and as time goes on, its cumulative error becomes progressively larger and gradually moves away from the true trajectory in the end, i.e., the situation is worse on the left-hand side and relatively better on the remaining three sides. The trajectory of the fused positioning algorithm is close to the real trajectory on the whole. Figure 16 shows the CDF plots of the two localization methods.

The VIO localization error in this experiment does not show a sudden change in error compared to experiment (B), but rather a smoother gradual increase, but the maximum localization error reached a value of about 4 m due to the large area. It can be seen that the accumulation of errors in the large area is an urgent problem for VIO position estimation, irrespective of the problem of light and darkness. With fusion, the overall localization errors of our algorithm were all within 2 m, with most of the points having a localization error within 1 m. Table 4 shows a comparison of the localization errors of the two algorithms. After fusion, the average localization error of our algorithm was 0.72 m, which is about a 67.2% improvement over the single VINS-Mono [25] localization method. Therefore, our proposed fusion positioning algorithm can reduce the accumulation of errors and improve indoor positioning accuracy.

## 5. Conclusions

VIO localization techniques have high accuracy over short periods of time, but are subject to cumulative errors and cannot fulfill the positioning needs of indoor robots for long periods of time. Global localization techniques that rely on indoor signal sources do not suffer from cumulative errors, but the overall localization accuracy is low and cannot provide reliable global localization results. In this paper, the 5G and geomagnetic signals are first fused for global localization, and then the global localization results are fused with the localization results by the ES-EKF algorithm. Through experiments with unmanned vehicle carriers, the following conclusions can be drawn:

(a) The BP neural network-based fusion positioning of 5G and geomagnetic signals reduces the large positioning errors from a single source. After fusion, the average positioning error is 0.49 m, which is 0.27% and 0.17% better than the single sensor positioning methods of geomagnetic and 5G signals, respectively;

(b) Loosely combining the local localization results of VIO with the global localization results of 5G and geomagnetic, not only solves the problem of VIO not being able to provide global localization results, but also reduces the positioning error of VIO. After loosely coupling, the average localization error in the two scenarios is 0.61 m and 0.72 m, which is a 69.0% and 67.2% improvement, respectively, on the currently popular VINS-Mono algorithm with a single VIO. This demonstrates that the algorithm in this paper can improve the accuracy and robustness of indoor positioning.

Considering that the filter-based method proposed in this paper can provide real-time localization results, but still lacks in accuracy compared to the optimization-based method, in future work we will consider how to combine both real-time and accuracy of localization. In addition, given that the localization work in this paper was mainly carried out on a two-dimensional plane, with the future development of space technology, the 3D position information of mobile robots has also become more important, so in future research, we will consider using UAVs as platforms to achieve three-dimensional localization of robots.

## Figures and Tables

**Figure 1 sensors-23-01311-f001:**
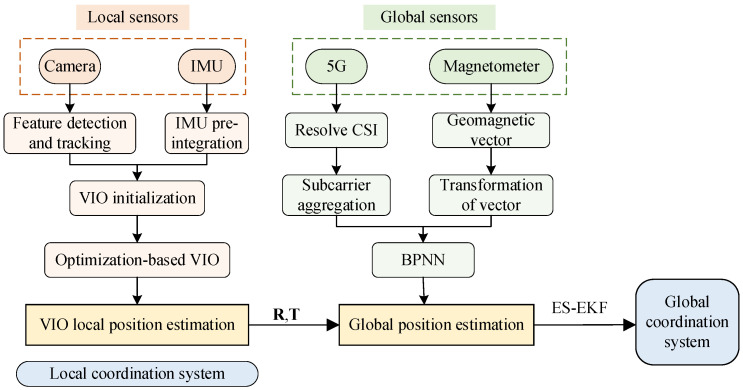
The architecture of the proposed system. **R** represents the rotation matrix converted from local coordinates to global coordinates and **T** is the translation matrix.

**Figure 2 sensors-23-01311-f002:**
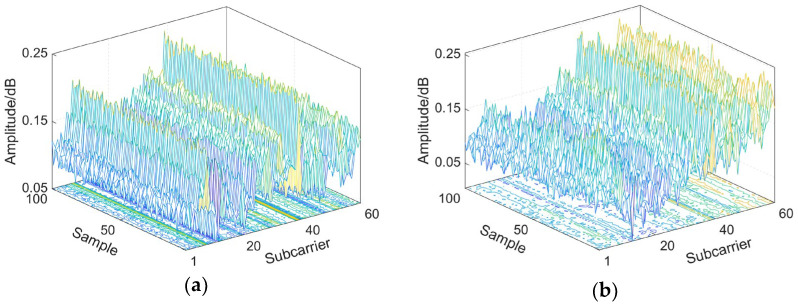
3D images of CSI data collected at different locations. (**a**) Location A. (**b**) Location B.

**Figure 3 sensors-23-01311-f003:**
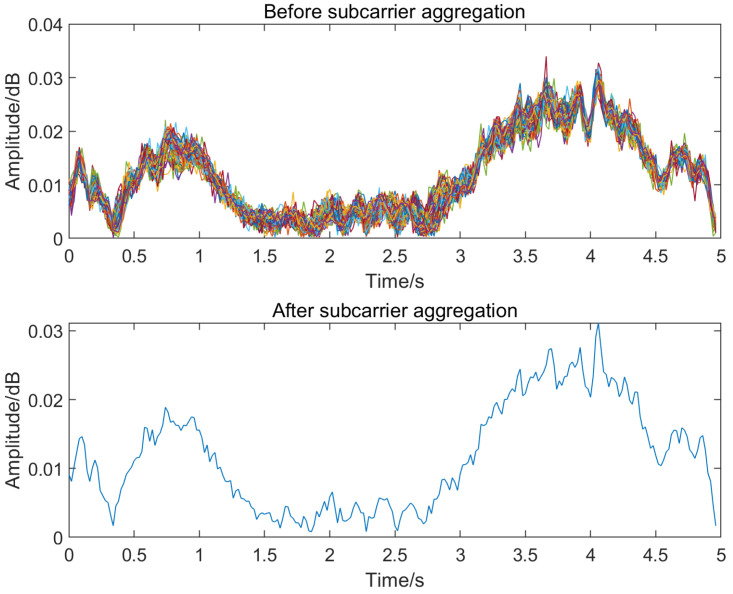
Amplitude comparison before and after subcarrier aggregation.

**Figure 4 sensors-23-01311-f004:**
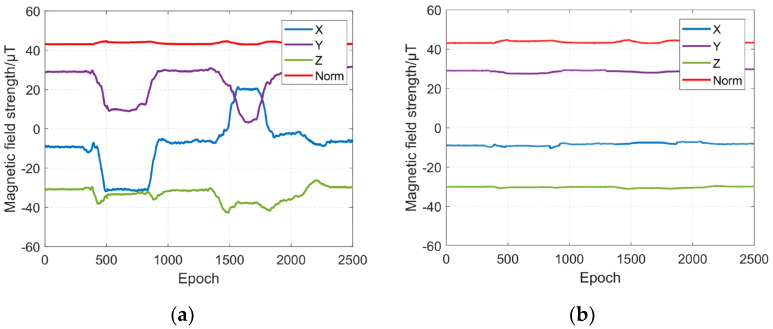
Magnetic field strength before (**a**) and after (**b**) the change.

**Figure 5 sensors-23-01311-f005:**
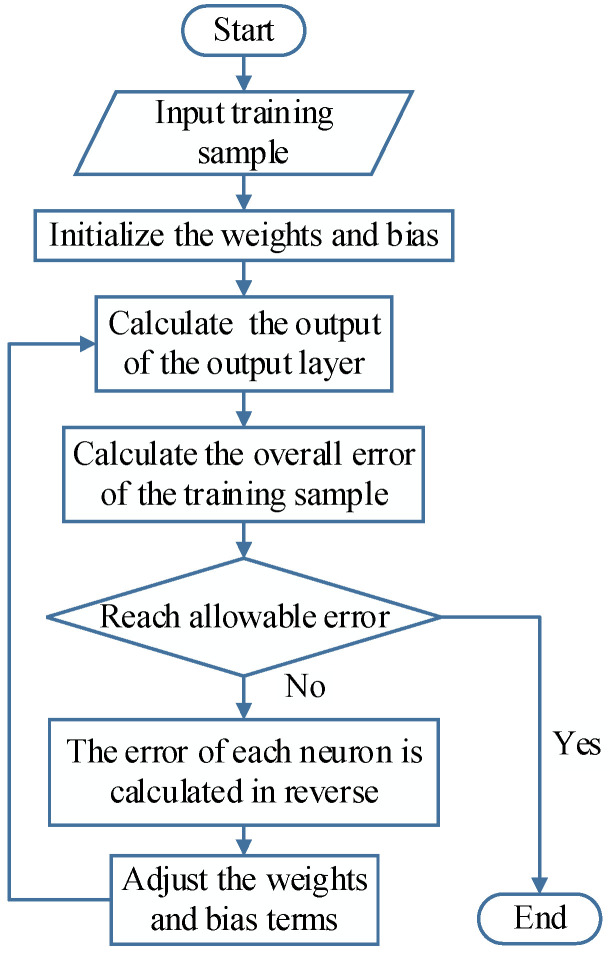
BP neural network training process.

**Figure 6 sensors-23-01311-f006:**
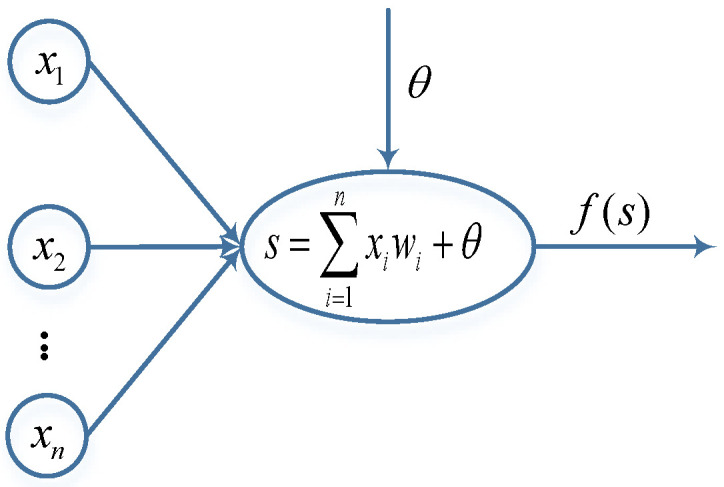
Basic structure of neurons, x represents the input information from the previous layer neurons, ω is the weight, θ is the bias and f(s) is the output of the neuron.

**Figure 7 sensors-23-01311-f007:**
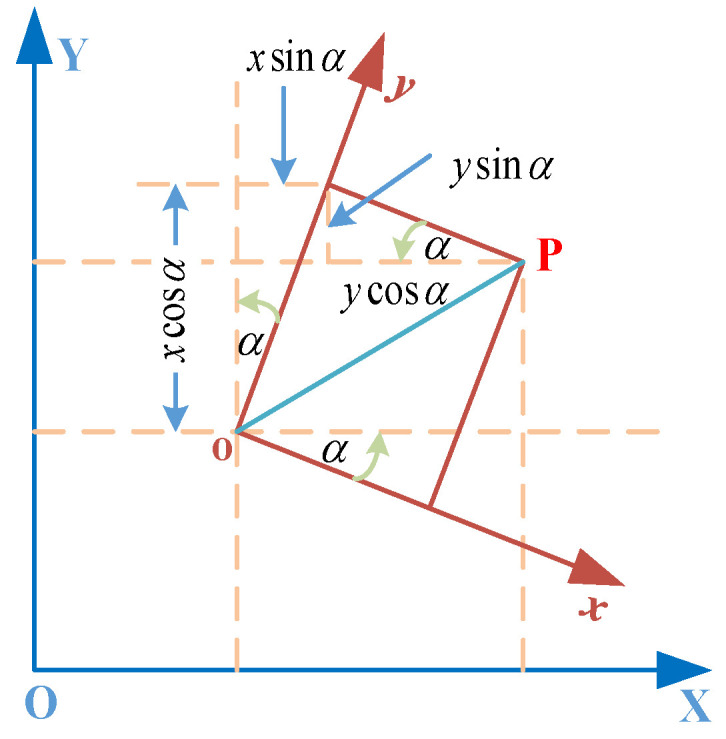
Coordinate transformation diagram, the point P is some point to be transformed, xoy is the local coordinate system, XOY is the global coordinate system and α is the angle of rotation.

**Figure 8 sensors-23-01311-f008:**
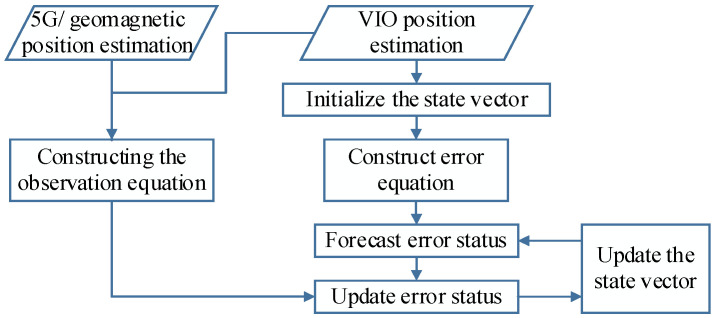
Combined positioning process.

**Figure 9 sensors-23-01311-f009:**
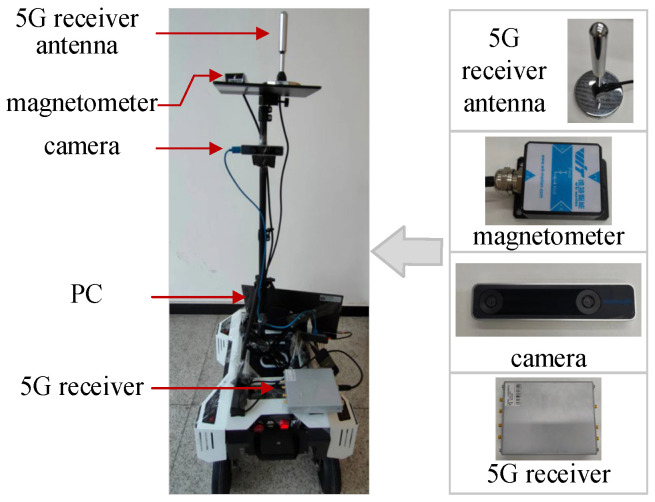
Experimental installation.

**Figure 10 sensors-23-01311-f010:**
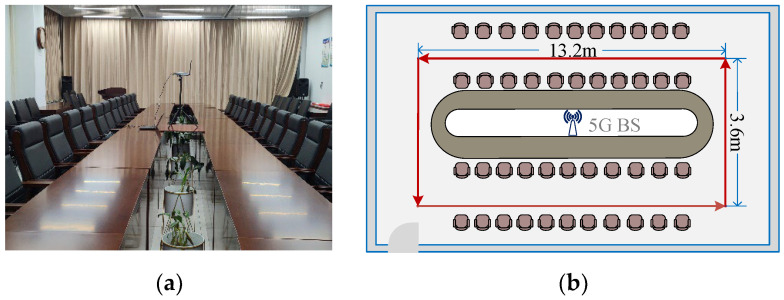
Conference room. (**a**) Scenario photograph. (**b**) Experimental approaches.

**Figure 11 sensors-23-01311-f011:**
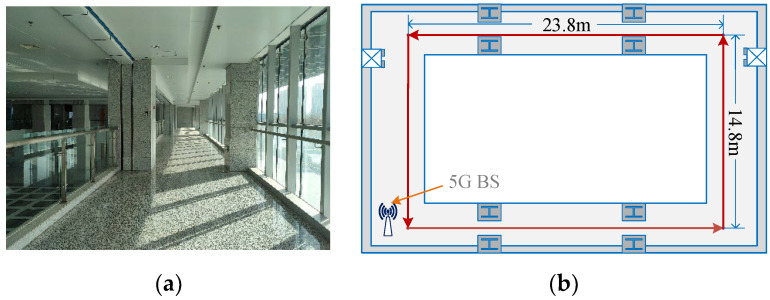
Covered corridor. (**a**) Scenario photograph. (**b**) Experimental approaches.

**Figure 12 sensors-23-01311-f012:**
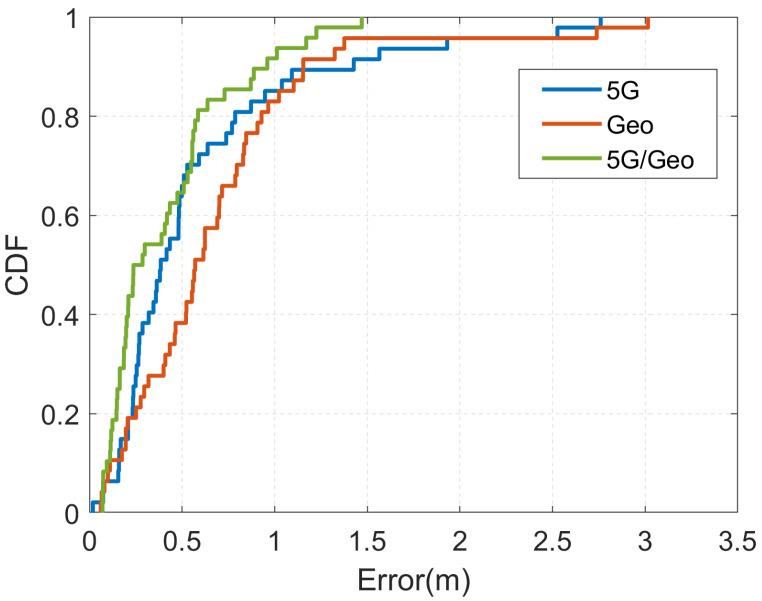
CDF of positioning errors with different method, blue is the 5G positioning result, red is the geomagnetic positioning result and green is the fusion positioning result.

**Figure 13 sensors-23-01311-f013:**
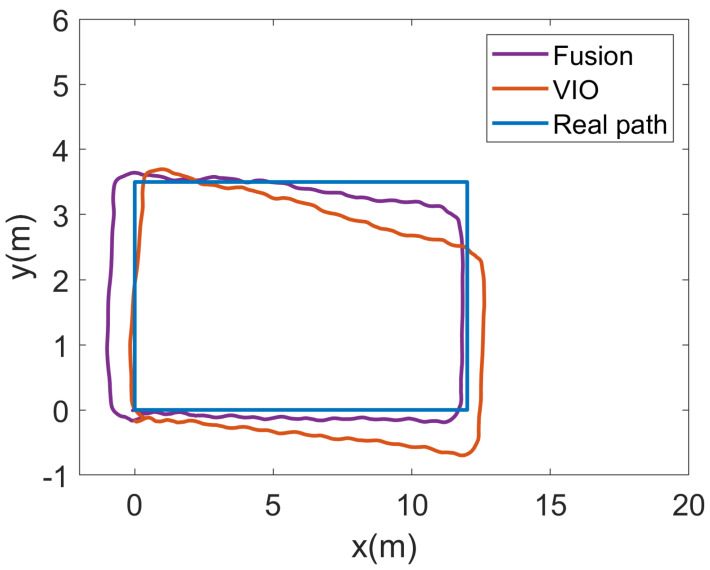
Moving trajectory of the conference room.

**Figure 14 sensors-23-01311-f014:**
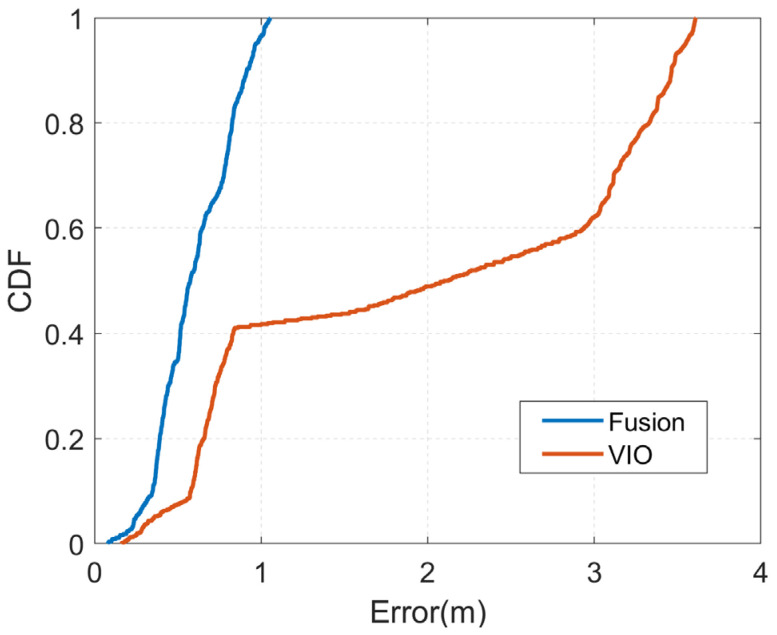
CDF of positioning errors with different method of the conference room.

**Figure 15 sensors-23-01311-f015:**
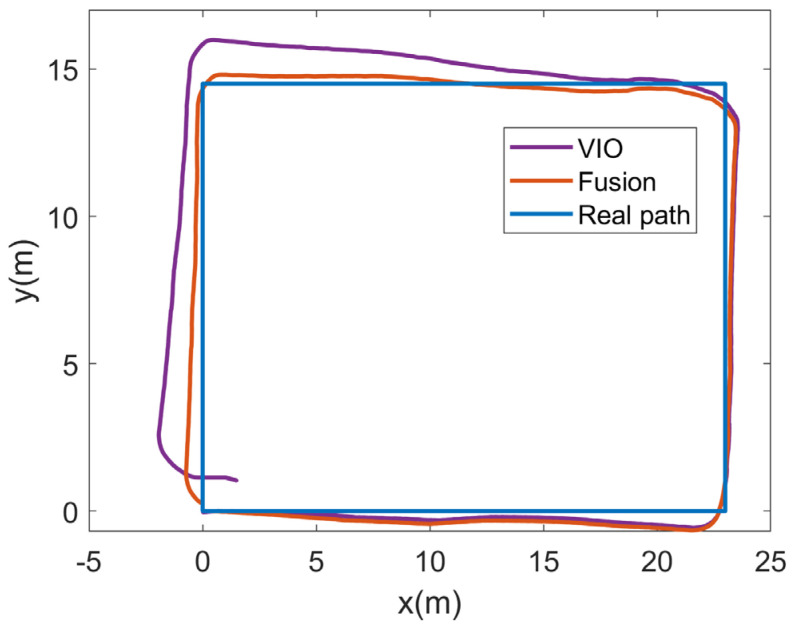
Moving trajectory of the covered corridor.

**Figure 16 sensors-23-01311-f016:**
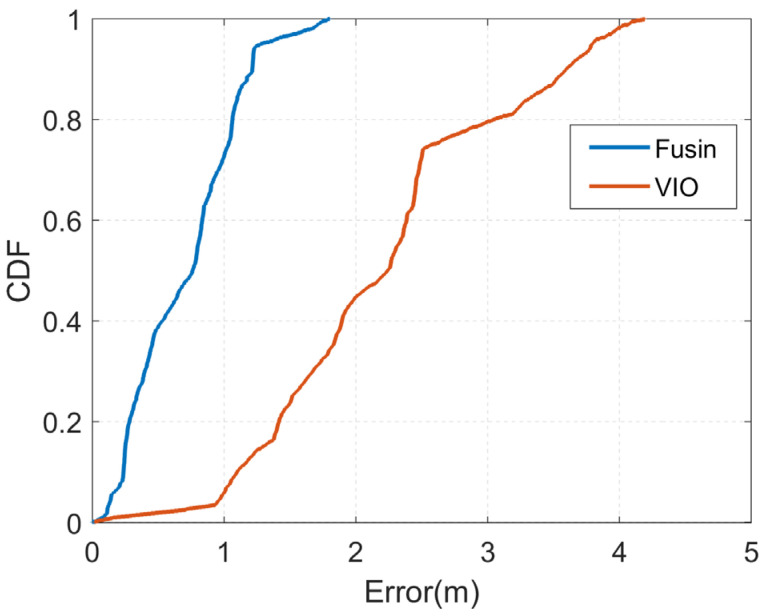
CDF of positioning errors with different method of the covered corridor.

**Table 1 sensors-23-01311-t001:** Experimental equipment parameters.

Name	Type	Frequency of Output (Hz)
Camera	RealsenceT265	Image: 50 IMU: 200
Magnetometer	Wit-motion HWT901B-232	50
5G receiver	5G experimental platform, Wuhan University	50

**Table 2 sensors-23-01311-t002:** Position error comparison.

Signal Source	Average (m)	Maximum (m)
5G	0.59	2.76
Geomagnetic matching	0.67	3.01
5G/geomagnetism matching	0.49	1.47

**Table 3 sensors-23-01311-t003:** Position error comparison of the conference room.

Position Technique	Average (m)	Maximum (m)
VIO	1.97	3.60
Fusion	0.61	1.05

**Table 4 sensors-23-01311-t004:** Position error comparison of the covered corridor.

Position Technique	Average (m)	Maximum (m)
VIO	2.21	4.19
Fusion	0.72	1.80

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
