# Peer review of "A Novel Deep Learning Approach to 5G CSI/Geomagnetism/VIO Fused Indoor Localization"

_sensors, 2023, doi:10.3390/s23031311_

Round 1

Reviewer 1 Report

The authors present an interesting paper about a novel indoor fingerprintig method based on 5G CSI/Geomagnetism/VIO techniques. However you should implement some improvemnts, namely:

1. Perform a checl of written english, in particular starting paragraphs with capital letters.

2. Authors should clearly identify that the method they developed and present is intended for indoor localization, and only for LOS scenarios.

3. Thus the test conditions should be better detailed! For example, part of the fusion model depends on the luminusity conditions and, therefore, the authors should not only refer, empirically, "better luminusity against worst luminusity conditions", which is the threshold of the process in lumens? How do the video camera features influence, or can influence the process?

4. Also the results could and deserved better explanation or reasoning! for instance, what is the reason, if any, for the presented model to present, in both experiments, better results in three of the faces/sides of the routes and in the same (left side), it presents worse results. According to Figures 13 e 15, there seems to be a pattern in this regard! Are they better lit areas? Is it related to placement, distance/diretivity, of the antenna, or due to the presence of obstacles that interrupt the LOS.

5. The analysis of the results, should be carried out in a more quantitative way, ie, how much better is the autors´model compared to those used as benchmarking. What is the matching poins percentage? Which situation  in the scenario contribute most to the increase in absolute and cumulative error.

Finally, congratulation for your work

Reviewer 2 Report

The authors proposed a fused 5G/geomag-netism/VIO indoor localization method. The paper is interesting and well-organized, but there are some comments and suggestions as follows:

- I suggest listing the paper's contribution in several points in the introduction section.

- You need to add more related works such as "An overview of indoor localization technologies: Toward IoT navigation services".

- The word "Where" after the given equation should be written in lowercase. 

- The path loss in equation 6 does not represent a 5G network  

- There are no details on the data collection, dataset size

- Figures 2,3,4,12,13,14,15 and 16 replot with high resolution. 

- The experimental testbed is small, thus, the localization error obviously small even if you use a basic localization algorithm. Moreover, you used complicated equipment. 

- The proposed work did not compare with any state-of-the-art works. 

- The conclusion should be rewritten and include the study outcome like accuracy, and localization error. 

Author Response

Please see the attachment:

Round 2

Reviewer 2 Report

Thanks for addressing my concerns, but there are some comments that need to be justified: I did not see details on the data collection and dataset size.  The comparison with any state-of-the-art works is should not be with the same technology (5G), but it can be with other technologies. There is a work that used the Bayesian network entitled" A three-dimensional pattern recognition localization system based on a Bayesian graphical model", but you did not include it in the related work. 
